# Optimization of seebeck coefficients in polyaniline-doped manganese dioxide nanocomposites

**Jay Molino**[1,2]*, **Muhammad Ibrahim**[3], **Rolando Serra**[4], **Svetlana de Tristán**[1]

**1** Universidad Especializada de las Américas (UDELAS), Faculty of Biosciences and Public Health, Biomedical Engineering, Centro I+D+i de Biotecnología, Energías Verdes y Cambio Climático, Albrook, Paseo de La Iguana, Republic of Panama, **2** Sistema Nacional de Investigación (SNI), SENACYT, Panama City, Republic of Panama, **3** Faculty of Engineering and Science, Bahauddin Zakariya University – BZU, Punjab, Pakistan, **4** Departamento de Física, Universidad Tecnológica de La Habana José Antonio Echeverría, La Habana, Cuba

* jay.molino@udelas.ac.pa

## Abstract

Polyaniline (PANI) and PANI-$MnO_2$ composites were synthesized via a chemical route with varying manganese dioxide ($MnO_2$) content, specifically 5wt% and 15wt%. X-ray diffraction (XRD) confirmed the structural formation of both PANI and PANI-$MnO_2$ composites. The direct current conductivity was measured, showing an increase with temperature: at 393K, pure PANI had a conductivity of $2.25 \times 10^{-4}$ S/cm, which increased significantly in the composites, reaching $9.03 \times 10^{-4}$ S/cm for the 15wt% $MnO_2$ composite. The Seebeck coefficient also increased with temperature and $MnO_2$ concentration, achieving a maximum value of 52 mV $K^{-1}$ at 373K for the 15wt% $MnO_2$ composite. These results indicate that the synthesized PANI- $MnO_2$ composites exhibit semiconducting behavior with improved thermoelectric properties, making them promising candidates for applications in thermoelectric devices such as generators and thermopiles. The study highlights the potential of these materials in enhancing the efficiency of thermoelectric energy conversion.

## Introduction

The study of thermoelectric materials has gained significant traction due to their potential to convert waste heat into useful electrical energy, addressing global energy concerns and contributing to sustainability [1,2]. Thermoelectric devices rely on the Seebeck effect for energy conversion, which requires materials with a high Seebeck coefficient, excellent electrical conductivity, and low thermal conductivity to achieve high efficiency. Polyaniline (PANI)-doped manganese dioxide ($MnO_2$) composites have potential to optimize these properties for thermoelectric applications. Recent advances have demonstrated innovative applications of the Seebeck effect, such as in self-powered temperature sensors that leverage photothermal-thermoelectric coupling for immunoassays, underscoring the broad utility of materials that exhibit enhanced thermoelectric performance [3,4]. Conducting polymers, such as polyaniline (PANI), have emerged as promising candidates for thermoelectric applications due to their tunable electrical properties, ease of synthesis, and environmental stability [5,6].

**Data availability statement:** All the data can be found in the main text and Supporting information.

**Funding:** The author(s) received no specific funding for this work.

**Competing interests:** The authors have declared that no competing interests exist.

Polyaniline (PANI) is a conducting polymer known for its unique electrical properties and ability to exist in various oxidation states, controllable through doping [7,8]. The emeraldine form of polyaniline (PANI) exhibits significant conductivity when doped with protonic acids, as protonation enhances charge carrier mobility and interchain hopping. Recent studies demonstrate that acid doping creates ordered polaronic structures, improving electrical properties [9–11].

Manganese dioxide ($MnO_2$), an inorganic compound with a high Seebeck coefficient, has been widely studied for its potential in energy storage and conversion devices [12,13]. $MnO_2$ has several polymorphic forms, each with distinct structural and electrochemical properties [14]. Combining $MnO_2$ with PANI can enhance the thermoelectric properties of the composite material, leveraging the high Seebeck coefficient of $MnO_2$ and the conductive nature of PANI [15,16].

This study synthesized PANI and PANI- MnO2 composites with varying MnO2 content (5wt% and 15wt%). The structural properties of these composites were confirmed using X-ray diffraction (XRD). The D.C. conductivity was measured as a function of temperature, and the Seebeck coefficient was evaluated to determine the thermoelectric potential of the composites. This research aims to optimize the Seebeck coefficient in PANI-$MnO_2$ nanocomposites and assess their suitability for thermoelectric applications.

Doping PANI with different materials, such as $MnO_2$, can significantly influence its electrical conductivity and thermoelectric properties [17]. By optimizing doping levels and composite structures, it is possible to enhance the Seebeck coefficient, making PANI-$MnO_2$ composites highly efficient for thermoelectric applications, which involves maintaining high electrical conductivity while minimizing thermal conductivity, thus maximizing thermoelectric efficiency [18].

This research focuses on developing and characterizing conducting polymers and transition metal oxides, exploring new avenues for their application in thermoelectric devices. By enhancing the thermoelectric performance of PANI through $MnO_2$ doping, we aim to develop efficient, environmentally friendly materials for energy conversion technologies. The findings of this study have the potential to impact the design and optimization of next-generation thermoelectric devices, which are critical for improving energy efficiency in various industrial processes [19,20].

## Materials and methods

PANI was synthesized by mixing ammonium persulfate (APS) with aniline in an acidic medium, followed by filtration, washing, drying, and grinding. HCl-doped PANI and PANI-$MnO_2$ composites (5% and 15% $MnO_2$) were prepared using a similar method, adjusting $MnO_2$ concentrations. XRD confirmed structural properties. Electrical properties were measured using a Keithley 2400 SourceMeter. Seebeck coefficients were determined using a custom setup with a nano-voltmeter, measuring from 313K to 373K. The preparation of PANI and PANI-MnO₂ nanocomposites was conducted at controlled conditions. The polymerization was carried out at room temperature (~25°C), followed by vacuum drying at 90°C to ensure uniformity and removal of residual moisture.

### Synthesis of pure PANI

Polyaniline (PANI) was synthesized by dissolving 12.25 g of APS in 50 mL of distilled water, stirring continuously. Separately, 50 mL of deionized water was mixed with 5 mL of aniline for 30 minutes. The APS solution was added dropwise to the aniline solution, followed by 5 mL of hydrochloric acid (HCl), maintaining a neutral pH. The reaction mixture was stirred for

30–60 minutes and left in the dark for 24 hours. The resulting PANI precipitate was filtered, washed with deionized water, dried at 60°C, and ground into a fine powder.

### Synthesis of HCl-doped PANI

HCl-doped PANI was prepared by dissolving 12.25 g of APS in 50 mL of HCl and stirring. Separately, 50 mL of HCl was mixed with 5 mL of aniline for 30 minutes. The APS solution was added dropwise to the aniline solution, followed by 5 mL of HCl, maintaining a neutral pH. The mixture was stirred for 30–60 minutes and left in the dark for 24 hours. The resulting precipitate was filtered, washed with deionized water, dried at 60°C, and ground into a fine powder.

### Synthesis of HCl-doped polyaniline- $MnO_2$ composites

HCl-doped PANI- $MnO_2$ composites were synthesized by preparing two APS solutions in 50 mL of HCl with 12.25 g of APS. For the 5% MnO2 composite, 0.25 g of $MnO_2$ was added to 50 mL of HCl, stirred for 30 minutes, followed by 5 mL of aniline. The mixture was combined with the APS solution, stirred, left in the dark for 24 hours, filtered, dried at 90°C, and ground. The same procedure was followed for the 15% $MnO_2$ composite, using 0.75 g of $MnO_2$.

### Characterization

Structural properties were analyzed using a Bruker D8 Advance diffractometer with Cu Kα radiation (λ = 1.5406 Å), scanning from 10° to 80° at a rate of 2° per minute. Conductivity was measured using a Keithley 2400 SourceMeter in the temperature range of 313K to 393K with a two-probe method. The Seebeck coefficient was measured using a custom setup with a temperature control system and a nano-voltmeter (2182A KEITHLEY). Thin films for the Seebeck coefficient measurements were prepared by the doctor-blade method, and measurements were taken from 313K to 373K with a 2 cm distance between contact points.

## Results

### X-ray diffraction (XRD) analysis

The XRD patterns for pure polyaniline, HCl-doped polyaniline, HCl-doped polyaniline with 5wt% $MnO_2$, and HCl-doped polyaniline with 15wt% $MnO_2$ are presented in Fig 1. The XRD pattern for pure polyaniline (PANI) exhibits crystalline peaks ranging from 2θ = 9.5° to 2θ = 29.9°, with prominent peaks at 2θ = 20.77° and 26.63° corresponding to the (020) and (200) crystallographic planes, respectively, confirming the material's primarily amorphous nature with some crystalline features. These findings are consistent with reported literature on polyaniline [21,22]. The amorphous characteristics of PANI contribute to its versatility in functional materials and applications, such as biosensors and electrochromic devices. Recent studies have also explored such properties, including the bioinspired and multiscale hierarchical design of flexible pressure sensors for high-throughput detection, showcasing polyaniline's versatility in advanced applications [23]. The broad peak at 26.63° is consistent with reported data, demonstrating the characteristic broad peak of polyaniline (Fig 1A).

The XRD pattern also shows partially crystalline peaks for HCl-doped polyaniline compared to pure PANI. Broad peaks at 2θ = 14.57° and 20.22° and a semi-crystalline peak at 2θ = 25.23° with a d-value of 2.98 Å indicate a slight crystalline nature due to π-conjugation in polyaniline (Fig 1B).

Comparing the XRD patterns of pure polyaniline and HCl-doped polyaniline reveals a hump peak at 2θ = 25.23° in the HCl-doped sample, indicating a partially crystalline nature.

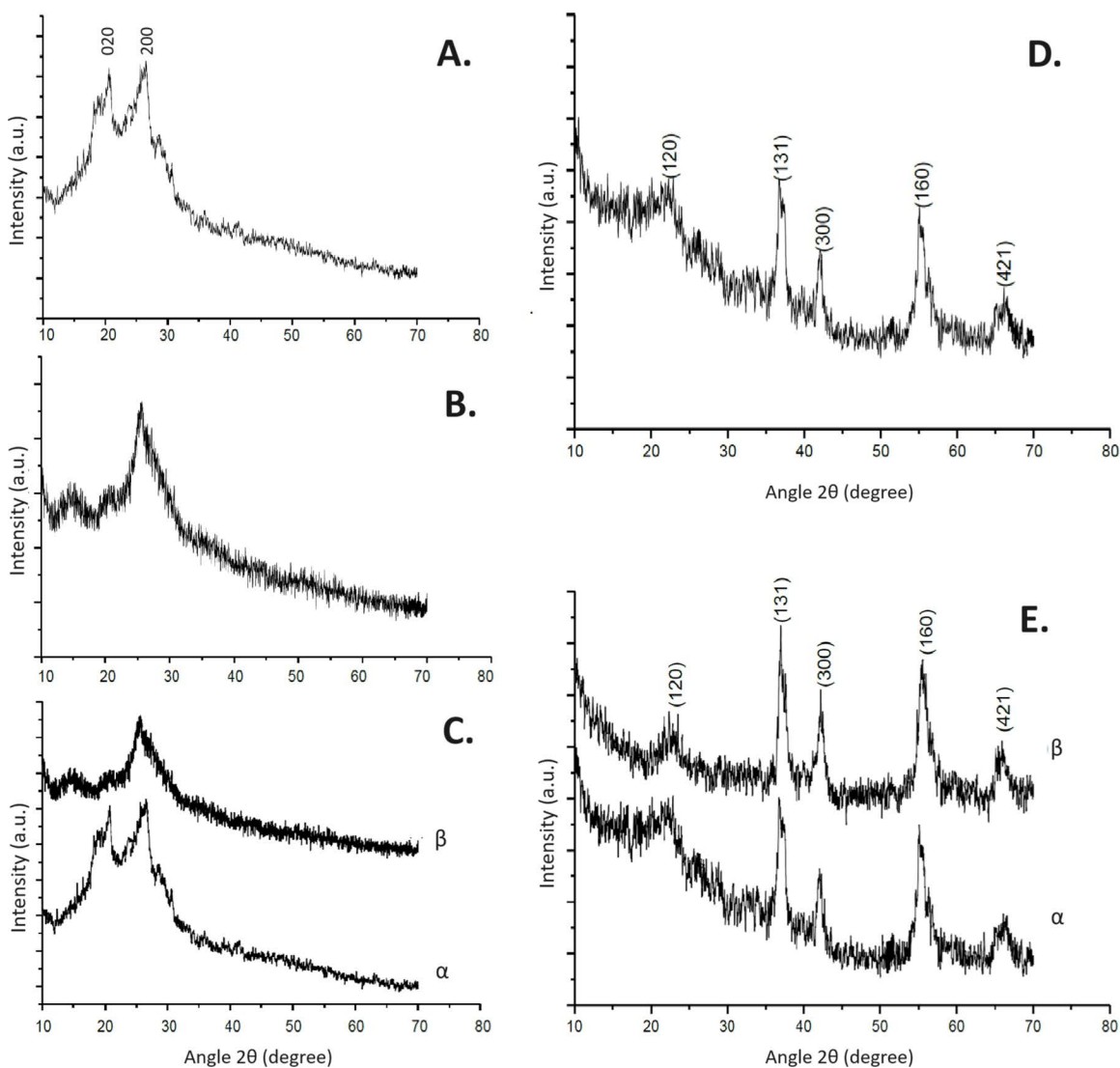

**Fig 1.** *Panel A.* X-ray diffraction of pure polyaniline; *Panel B.* XRD of Polyaniline doped with HCl; *Panel C.* XRD of (α) Pure polyaniline (β) HCl doped polyaniline; *Panel D.* XRD of MnO2; XRD of (α) HCl doped Polyaniline-5wt% MnO2 (β) HCl doped Polyaniline-15wt% MnO2.

This semi-crystalline arrangement suggests an interaction mechanism between the polymer particles and the dopant (Fig 1C). The XRD pattern of pure $MnO_2$ shows well-defined peaks at angles $2\theta$ = 22.29°, 37.2°, 42.2°, 55.3°, and 66.63°, corresponding to the (120), (131), (300), (160), and (421) planes, respectively. The high-intensity peak at $2\theta$ = 37.2° with a d-value of 2.38 Å confirms the crystalline nature of $MnO_2$. The average grain size, calculated using Scherer's formula, was determined to be 55 nm (Fig 1D).

The XRD patterns of HCl-doped polyaniline-$MnO_2$ composites with 5wt% and 15wt% $MnO_2$ show partially sharp peaks of $MnO_2$ integrated with HCl-doped polyaniline. The crystallinity of $MnO_2$ within the polymer matrix increases with $MnO_2$ concentration, enhancing the conductive nature and confirming the homogeneous mixing of $MnO_2$ within the polyaniline chain (Fig 1E).

The detailed results from the XRD analysis confirm the successful synthesis and incorporation of MnO2 into the polyaniline matrix, enhancing the structural and crystalline properties of the composites. The observed peaks are consistent with the reported values in the literature, validating the synthesis process and the composite's crystalline nature.

The XRD analysis confirmed increased crystallinity in $MnO_2$-doped PANI composites. This structural reorganization can enhance charge carrier mobility by reducing scattering and improving conductivity pathways. To evaluate this effect, D.C. electrical conductivity measurements were performed.

## D.C. electrical conductivity measurements

The D.C. electrical conductivity of all prepared samples was measured using a two-probe method, applying a voltage range of 20V. Conducting polymers, when doped, exhibit a range of electrical conductivities from semiconducting to metallic. The conductivity of these materials also depends on temperature, generally increasing with rising temperature, indicative of semiconducting behavior. Polyaniline (PANI), a prominent conducting polymer, demonstrates this temperature-dependent conductivity.

The I-V characteristics of pure PANI were studied at various temperatures (313K-393K). The I-V curves indicate that the resistivity of pure PANI is higher than that of HCl-doped PANI and its composites. Consequently, the D.C. conductivity of pure PANI is lower. As temperature increases, the conductivity of pure PANI also increases (Fig 2A).

The I-V characteristics of HCl-doped PANI were evaluated at various temperatures (313K-393K). The resistance (R) was calculated using Ohm's law and converted to conductivity ($\sigma$) using the equation $\sigma = L/RA$. The resistivity of HCl-doped PANI is lower than that of pure PANI, resulting in higher conductivity, which increases with temperature, as shown in Fig 2B.

The electrical properties of HCl-doped PANI-5wt% $MnO_2$ were observed at temperatures ranging from 313K to 393K. The I-V curves were analyzed to calculate resistance and conductivity. The resistivity of this composite is lower than that of pure PANI, indicating higher conductivity, as shown in Fig 2C.

Similarly, the I-V characteristics of HCl-doped PANI-15wt% $MnO_2$ were measured. The resistivity of this composite is lower than that of pure PANI and decreases with increasing $MnO_2$ content, resulting in higher conductivity, as depicted in Fig 2D.

## D.C. conductivity and seebeck coefficient analysis of HCl-doped PANI and HCl-$MnO_2$ composites

The D.C. conductivity of PANI-doped with HCl and its $MnO_2$ composites was studied as a function of temperature. The results show that conductivity increases with temperature and MnO2 content, as illustrated in Fig 3A. MnO2, an n-type semiconductor with a large band gap, enhances carrier concentration and conductivity.

The activation energy for these composites was calculated using temperature-dependent D.C. conductivity data. By applying a linear fit to $\ln(\sigma)$ vs. $1/T$, the activation energy was determined and found to decrease with increasing $MnO_2$ content (Fig 3B). This reduction in activation energy signifies enhanced charge carrier mobility and reduced energy barriers, which is further evidenced by Fig 3C, showing a significant decrease in activation energy with higher MnO2 content. These findings confirm that the HCl-doped PANI-$MnO_2$ composites exhibit improved conductivity with increasing temperature and $MnO_2$ content, consistent with the behavior of semiconducting materials where higher temperatures facilitate better charge carrier movement.

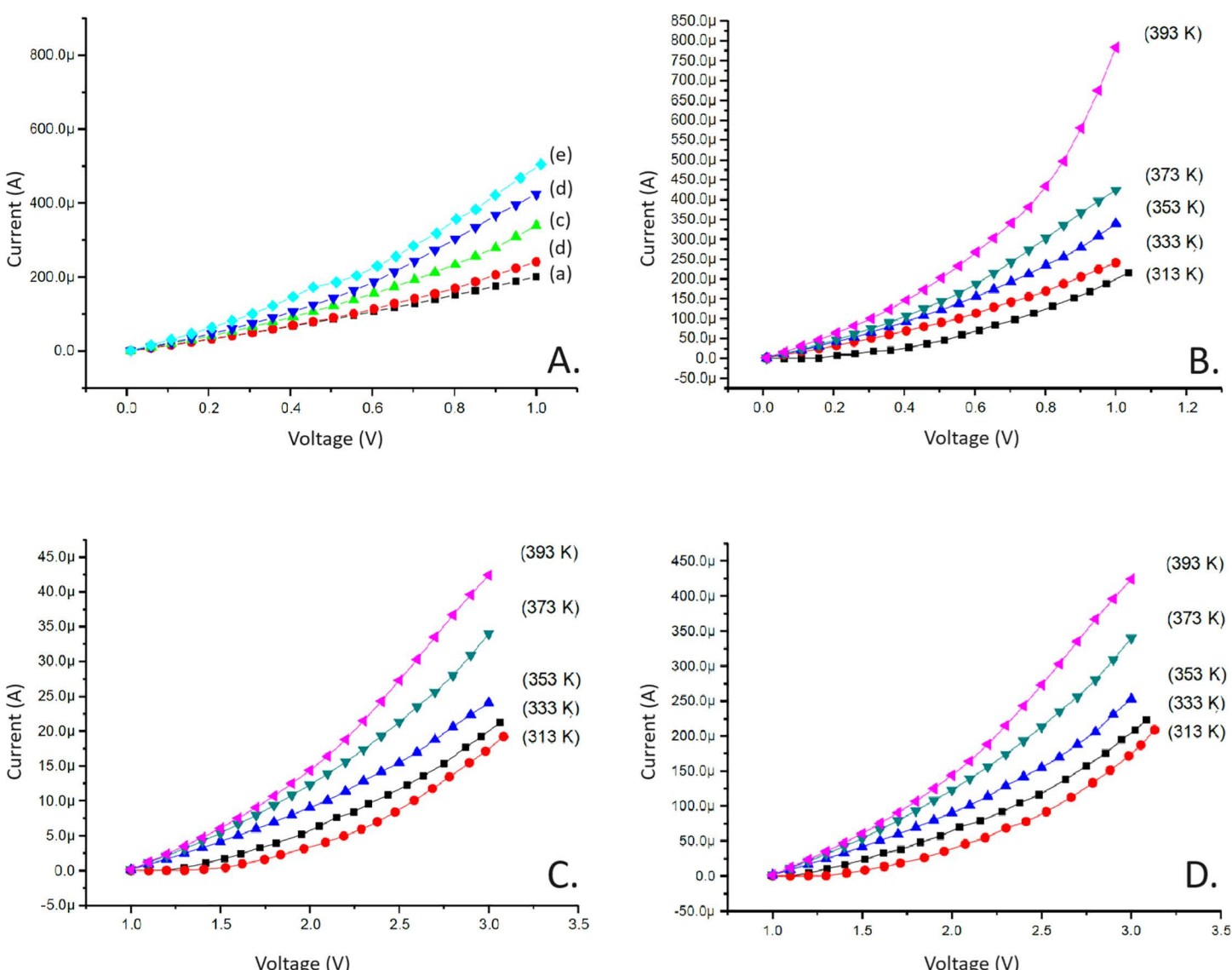

**Fig 2.** *Panel A.* **Characteristic I-V curve of Pure PANI at various temperatures (a) 313K (b) 333K (c) 353K (d) 373K (e) 393K.** *Panel B.* Characteristic I-V curve of HCl doped PANI at various temperatures (a) 313K (b) 333K (c) 353K (d) 373K (e) 393K. *Panel C.* Characteristic I-V curve of PANI 5wt%-MnO$_2$ at various temperatures (a) 313K (b) 333K (c) 353K (d) 373K (e) 393K. *Panel D.* Characteristic I-V curve of PANI 15wt%-MnO2 at various temperatures (a) 313K (b) 333K (c) 353K (d) 373K (e) 393K.

The Seebeck coefficient of HCl-doped PANI also demonstrated a significant increase with temperature. Initially, at 0 µV/K at 313K, the Seebeck coefficient rose to 32 µV/K at 373K, illustrating the semiconducting behavior of HCl-doped PANI, where higher temperatures enhance charge carrier movement (Fig 3D).

For HCl-doped PANI-MnO2 composites, the Seebeck coefficient increased significantly with the addition of MnO2 nanoparticles. The Seebeck coefficient for the HCl-doped PANI-MnO2 composite increased from 32 µV/K at 313K to 52 µV/K at 373K, indicating enhanced thermoelectric performance due to improved charge carrier efficiency. This improvement is more pronounced in the composites compared to HCl-doped PANI alone, where the Seebeck coefficient ranged from 0 µV/Kto 32 mV/K, while for the composites it ranged from 20 µV/Kto 52 µV/Kover the temperature range of 313K to 373K. The increase in the Seebeck

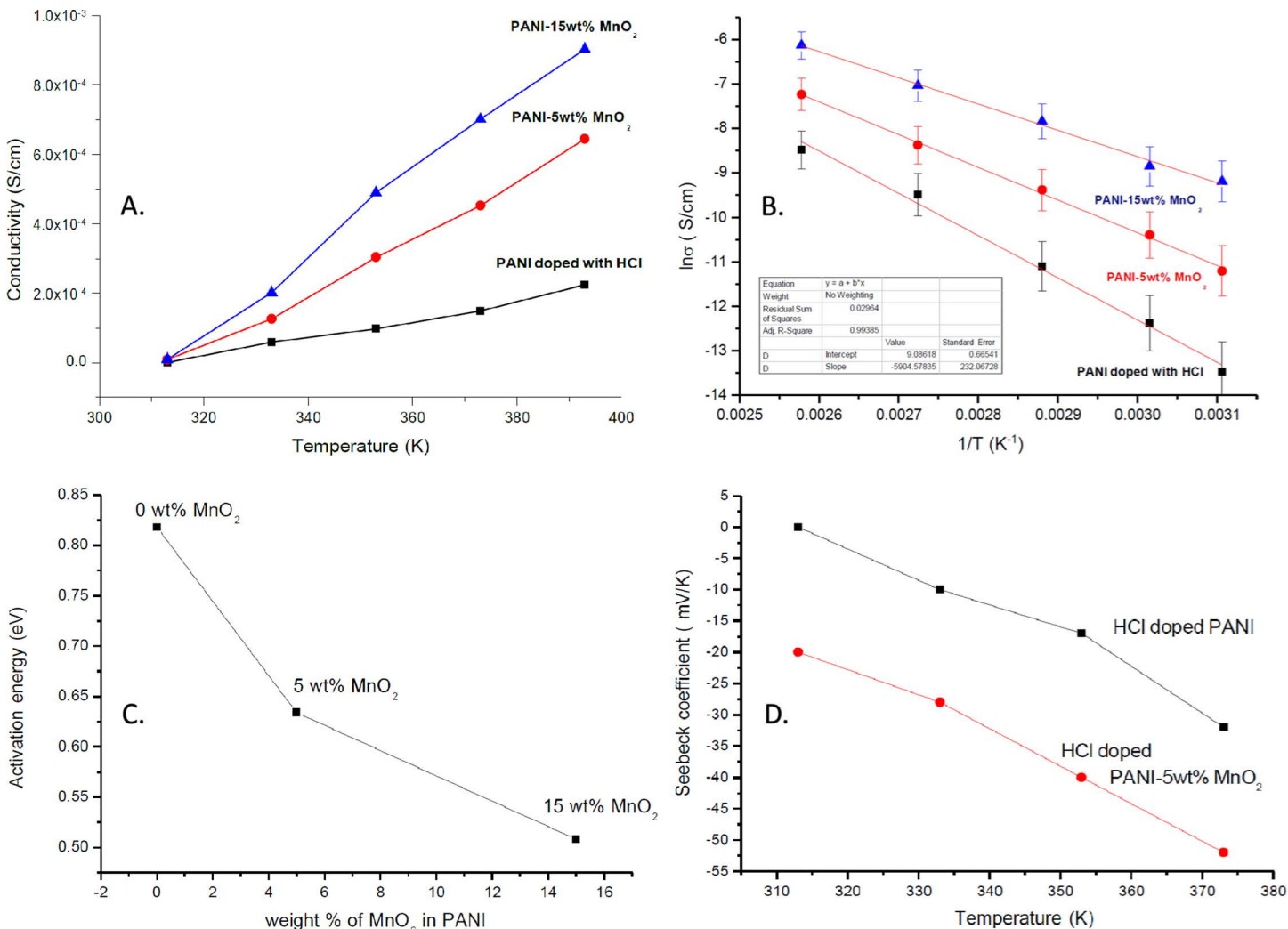

**Fig 3. *Panel 1*: D.C conductivity vs temperature dependent for (A) polyaniline doped with HCl (B) polyaniline doped with HCl-5wt% MnO₂ (C) polyaniline doped with HCl-15wt% MnO₂; *Panel 2*: ln(σ) vs 1/T for (A) polyaniline doped with HCl (B) polyaniline doped with HCl-5wt% MnO₂ (C) polyaniline doped with HCl-15wt% MnO₂; Panel 3: plot of Activation energy vs wt% of MnO₂ in PANI; Panel 4: Seebeck coefficient of PANI-doped with HCl and PANI with HCl-5wt% MnO₂.**

coefficient with temperature and MnO2 content is attributed to the enhanced efficiency of charge carriers and the generation of more Seebeck voltage.

## Discussion

The XRD analysis provided vital insights into the crystalline structure of the synthesized samples. PANI exhibited broad peaks at 2θ = 20.77° and 26.63°, corresponding to the (020) and (200) crystallographic planes, indicating their amorphous nature. Upon doping with HCl, PANI showed partially crystalline characteristics with peaks at 2θ = 14.57°, 20.22°, and 25.23°. This increase in crystallinity suggests enhanced alignment and ordering of the polymer chains due to HCl doping. The successful doping of MnO₂ into the PANI matrix was confirmed using XRD analysis, which showed characteristic peaks for MnO₂ and structural changes in PANI. While direct morphological evidence was not obtained (SEM, STEM, or EDAX), the

observed enhancements in electrical conductivity and Seebeck coefficient further support the formation of a nanoscale composite and the effective incorporation of $MnO_2$.

The XRD pattern of pure $MnO_2$ (Fig 1, Panel D) displays well-defined peaks at 2θ angles of 22.293°, 37.2°, 42.2°, 55.3°, and 66.6°, corresponding to the (120), (131), (300), (160), and (421) crystallographic planes, respectively. These peaks confirm the crystalline structure of $MnO_2$, consistent with previously reported studies. Recent works have highlighted the importance of $MnO_2$ in advanced applications. For example, $MnO_2$ nanosheets were utilized in a photoelectrochemical immunoassay for aflatoxin B1 detection, leveraging its etching reaction for signal transduction [24]. Another study demonstrated $MnO_2$ nanohybrids doped with oxygen and phosphorus for sensitive biomarker monitoring, emphasizing the material's versatility in photoelectrochemical sensing [25]. These findings support the significance of $MnO_2$'s crystalline structure in advanced sensing and energy applications.

The D.C. electrical conductivity measurements demonstrated an explicit dependency on temperature and dopant concentration. Pure PANI exhibited the lowest conductivity, increasing substantially upon HCl doping. This enhancement in conductivity is attributed to the introduction of charge carriers and improved polymer chain ordering due to HCl doping. Adding $MnO_2$ to the HCl-doped PANI further increased the conductivity, with the HCl-doped PANI-15 wt% $MnO_2$ composite showing the highest conductivity, which indicates that $MnO_2$ effectively enhances the charge transport properties of PANI. The conductivity values ranged from $2.25 \times 10^{-4}$ S/cm for pure PANI to $9.03 \times 10^{-4}$ S/cm for the 15 wt% $MnO_2$ composite at 393K. The significant increase in conductivity is likely due to the synergistic effect of $MnO_2$ and HCl doping, which improves the electronic structure and reduces the energy barrier for charge transport.

Seebeck coefficient measurements revealed positive values for all samples, characteristic of p-type semiconductors. The Seebeck coefficient increased with temperature and $MnO_2$ content, reflecting improved thermoelectric performance. The highest Seebeck coefficient was observed for the HCl-doped PANI-15 wt% $MnO_2$ composite, with a value of 52 μV/K at 373K, compared to 32 μV/K for HCl-doped PANI. This enhancement is attributed to the increased carrier concentration and improved mobility within the composite. The presence of $MnO_2$, with its wide band gap, helps increase the energy of charge carriers above the Fermi level, thereby enhancing the Seebeck coefficient.

The activation energy for electrical conduction decreased with increasing $MnO_2$ content. Pure PANI had the highest activation energy of 0.818 eV, which decreased to 0.508 eV for the HCl-doped PANI-15 wt% $MnO_2$ composite. This reduction in activation energy indicates that MnO2 doping facilitates more effortless charge carrier movement, enhancing electrical conductivity and thermoelectric properties. The decrease in activation energy with higher MnO2 content suggests that MnO2 acts as an efficient dopant, reducing the energy barriers within the polymer matrix and improving the overall charge transport properties.

Polyaniline (PANI)-doped manganese dioxide ($MnO_2$) nanocomposites offer several advantages and disadvantages in thermoelectric applications. These composites benefit from the synergistic combination of PANI's high electrical conductivity and $MnO_2$'s ability to enhance the Seebeck coefficient. Interestingly, the range of Seebeck coefficient improvement was similar (32 μV/K) for both systems, regardless of the $MnO_2$ concentration. This suggests that the energy filtering effects govern the enhancement, while the PANI matrix improves charge carrier mobility. Recent studies confirm that such composites show superior thermoelectric and thermal stability properties [4]. The inherent charge transport limitations within the PANI matrix also contribute to the observed plateau in Seebeck coefficient enhancement. These results indicate that optimizing the uniformity and interfacial interactions of $MnO_2$ within the polymer matrix could further improve thermoelectric performance.

MnO$_2$ also provides structural robustness and contributes to thermal stability, while PANI ensures environmental durability and processability, making the material suitable for long-term energy applications [26]. However, the low intrinsic electrical conductivity of MnO$_2$ can limit the overall performance if doping ratios are not optimized, and achieving homogeneous dispersion of MnO$_2$ within the PANI matrix requires advanced synthesis techniques. Despite these challenges, the PANI-MnO$_2$ nanocomposites demonstrate great potential for addressing critical needs in modern thermoelectric devices [27].

The current-voltage (I-V) characteristics presented in Fig 2B, 2C, and 2D demonstrate the electrical behavior of HCl-doped PANI, 5wt% MnO$_2$-PANI, and 15wt% MnO$_2$-PANI composites across different temperatures (313–393 K). As shown in Fig 2B, the HCl-doped PANI exhibits lower current values, indicating higher resistance than the MnO$_2$-doped composites. In contrast, adding MnO$_2$ results in enhanced electrical conductivity due to improved charge transport facilitated by the interaction between MnO$_2$ and the PANI matrix.

A closer comparison of the 5wt% and 15wt% MnO$_2$-PANI composites (Figs 2C and 2D) reveals that the 15wt% MnO$_2$ composite achieves significantly higher current values at elevated temperatures. The aforementioned is attributed to the increased MnO$_2$ content, which enhances carrier mobility and decreases the resistance of the composite material. However, at lower temperatures, the differences in current values between the 5wt% and 15wt% composites are less pronounced, suggesting that improved conductivity due to MnO$_2$ doping is more effective at higher thermal energy levels.

Quantitatively, the resistance of HCl-doped PANI decreases with temperature due to the thermal activation of charge carriers. Still, its conductivity remains lower than that of MnO$_2$-PANI composites across all tested conditions[28–30]. For the MnO$_2$-PANI composites, the observed trends in conductivity enhancement align with the Seebeck coefficient improvements, confirming the synergistic role of MnO$_2$ in enhancing thermoelectric properties.

## Conclusion

This study successfully synthesized and characterized HCl-doped polyaniline (PANI) and PANI-MnO$_2$ composites using an in-situ molecular polymerization approach. X-ray diffraction patterns confirmed that manganese dioxide nanoparticles are partially crystalline, whereas pure PANI exhibits an amorphous nature. Peaks corresponding to MnO$_2$ at $2\theta =$ 22.3°, 37.2°, 42.2°, 55.3°, and 66.6° exhibit increased intensity and sharper profiles with higher MnO$_2$ content, confirming enhanced crystallinity. These peaks correspond to the (120), (131), (300), (160), and (421) planes, respectively, of MnO$_2$.

The D.C. conductivity measurements significantly increased with HCl doping and MnO$_2$ addition, confirming enhanced charge transport. The highest conductivity of $9.03 \times 10^{-4}$ S/cm was observed for the 15 wt% MnO$_2$ composite at 393K. Activation energy calculations revealed a reduction from 0.818 eV (pure PANI) to 0.508 eV (PANI-15 wt% MnO$_2$), indicating improved charge carrier mobility.

The Seebeck coefficient analysis showed that MnO$_2$ addition enhances thermoelectric properties. The PANI-15 wt% MnO$_2$ composite exhibited the highest Seebeck coefficient of 52 µV/K at 373K, suggesting improved energy filtering effects and charge carrier dynamics.

These findings demonstrate that HCl-doped PANI-MnO$_2$ composites are promising candidates for thermoelectric applications, balancing high electrical conductivity and enhanced Seebeck response. Future work should explore further optimization of doping levels, interfacial engineering, and advanced morphological characterization (TEM/FESEM) to enhance performance for energy conversion applications.

## Supporting information

**S1 File. Seebeck Measuring Instrument and Data Analysis.**
(DOCX)

**S2 File. Figures.**
(RAR)

## Author contributions

**Conceptualization:** Muhammad Ibrahim.

**Data curation:** Jay Molino.

**Formal analysis:** Jay Molino, Muhammad Ibrahim.

**Funding acquisition:** Muhammad Ibrahim.

**Investigation:** Muhammad Ibrahim.

**Methodology:** Jay Molino, Muhammad Ibrahim.

**Project administration:** Muhammad Ibrahim.

**Resources:** Jay Molino, Muhammad Ibrahim.

**Software:** Jay Molino, Muhammad Ibrahim.

**Supervision:** Muhammad Ibrahim, Rolando Serra.

**Validation:** Jay Molino, Muhammad Ibrahim.

**Visualization:** Jay Molino, Rolando Serra, Svetlana de Tristán.

**Writing – original draft:** Jay Molino, Rolando Serra, Svetlana de Tristán.

**Writing – review & editing:** Jay Molino, Rolando Serra, Svetlana de Tristán.

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
