## [Decision Letter · Decision Letter 0]

27 Aug 2024

PONE-D-24-32705Seebeck Coefficients Optimization in Doped Polyaniline-Manganese Dioxide NanocompositesPLOS ONE

Dear Dr. Molino,

Thank you for submitting your manuscript to PLOS ONE. After careful consideration, we feel that it has merit but does not fully meet PLOS ONE’s publication criteria as it currently stands. Therefore, we invite you to submit a revised version of the manuscript that addresses the points raised during the review process.

The manuscript requires improvements on the experimental section. There are several missing experimental data that are essential to clarify the work. Clearer and substantial elaborations for the experimental findings are required to justify the work. The elaborations must also be supported by relevant literature/published work.

We look forward to receiving your revised manuscript.

Kind regards,

Dr Irene Ling

Academic Editor

PLOS ONE

Journal Requirements:

2. Please include captions for your Supporting Information files at the end of your manuscript, and update any in-text citations to match accordingly. Please see our Supporting Information guidelines for more information: http://journals.plos.org/plosone/s/supporting-information .

Reviewers' comments:

Reviewer's Responses to Questions

**Comments to the Author**

1. Is the manuscript technically sound, and do the data support the conclusions?

Reviewer #1: Partly

Reviewer #2: Partly

Reviewer #3: Yes

Reviewer #4: Yes

2. Has the statistical analysis been performed appropriately and rigorously? 

Reviewer #1: Yes

Reviewer #2: I Don't Know

Reviewer #3: Yes

Reviewer #4: No

3. Have the authors made all data underlying the findings in their manuscript fully available?

Reviewer #1: Yes

Reviewer #2: Yes

Reviewer #3: Yes

Reviewer #4: Yes

4. Is the manuscript presented in an intelligible fashion and written in standard English?

Reviewer #1: No

Reviewer #2: No

Reviewer #3: Yes

Reviewer #4: Yes

5. Review Comments to the Author

Reviewer #1: This manuscript optimized Seebeck coefficients in polyaniline-doped manganese dioxide nanocomposites. Various manganese dioxide contents were doped with polyaniline by using a chemical route. The structural formation and the direct current conductivity were measured. Experimental results indicated that the Seebeck coefficients increased with the increment of temperature and MnO2 content, and exhibited semiconducting behaviors with improved thermoelectric properties for the fabrication of thermoelectric devices such as generators and thermopiles.

Specific comments:

1. Recently, different materials have been developed for the fabrication of thermoelectric devices. What are the advantages and disadvantages of polyaniline-doped manganese dioxide nanocomposites? Please specify them in the conclusion section.

2. The present title is somewhat confusing, and it should be reorganized. Suggested: Optimization of Seebeck coefficients in polyaniline-doped manganese dioxide nanocomposites.

3. Please remove the abbreviations in the keywords. Meanwhile, too many keywords are present, which should not exceed six keywords.

4. Thermoelectric devices rely on the Seebeck Effect and require materials with high Seebeck coefficients, excellent electrical conductivity, and low thermal conductivity for efficient energy conversion. One recent work (Self-powered temperature sensor with seebeck effect transduction for photothermal-thermoelectric coupled immunoassay) should be also mentioned for the Seebeck effect.

5. The XRD pattern for pure polyaniline (PANI) exhibits crystalline peaks ranging from 2θ = 9.5° to 2θ = 29.9°, with prominent peaks at 2θ = 20.77° and 26.63° corresponding to the (020) and (200) crystallographic planes, respectively, indicating its amorphous nature. Recent reports (Platinum Nanozyme-Catalyzed Gas Generation for Pressure-Based Bioassay Using Polyaniline Nanowires-Functionalized Graphene Oxide Framework; Pressure-based biosensor integrated with flexible pressure sensor and electrochromic device for visual detection) can be referred for polyaniline.

6. The XRD pattern of pure MnO2 shows well-defined peaks at angles 2θ = 22.29°, 37.2°, 42.2°, 55.3°, and 66.63°, corresponding to the (120), (131), (300), (160), and (421) planes, respectively. Some important researches (Signal-on photoelectrochemical immunoassay for aflatoxin B1 based on enzymatic product-etching MnO2 nanosheets for dissociation of carbon dots; Signal-on photoelectrochemical immunoassay mediated by the etching reaction of oxygen/phosphorus co-doped g-C3N4/AgBr/MnO2 nanohybrids) might be given on this topic.

7. The broad peak at 26.63° is consistent with reported data, demonstrating the characteristic broad peak of polyaniline. Relative report (Bioinspired and multiscale hierarchical design of a pressure sensor with high sensitivity and wide linearity range for high-throughput biodetection) can be provided for pure polyaniline.

8. Please carefully check all references, and typeset these references according to the guideline of this journal. Strangely, most references are before 10 years. Please update them! No references are given after 2020 year.

Reviewer #2: The authors reported Seebeck coefficient optimization in MnO2 doped PANI nanocomposites. In my opinion, this manuscript has low quality and insufficient to be published in this state. To be honest, I did not understand why MnO2 was used to dope PANI. MnO2 has very low electrical conductivity, however, MnO2 doped PANI shows increasing trend of electrical conductivity. In addition, is the unit of Seebeck coefficient really mV/K for PANI? except these things, this manuscript has lack of appropriate discussion and English is need to be improve.

Reviewer #3: Comments on the Research article entitled as “Seebeck Coefficients Optimization in Doped Polyaniline-Manganese Dioxide Nanocomposites” is given bellow:

The manuscript entitled as “Seebeck Coefficients Optimization in Doped Polyaniline-Manganese Dioxide Nanocomposites” aims to focus on the development of Polyaniline (PANI) and PANI-MnO2 composites and examined their semiconducting behavior. The work is interesting and novel. However the manuscript needs some minor improvement and after addressing that I recommend it to publish in “PLOS ONE”. Below are the some points that should be answered by the authors:

1. What was the temperature of preparation of PANI or PANI@MnO2 nanocomposites?

2. How could authors demand that the so produced PANI@MnO2 attained the nano-dimensions; have they taken TEM or FESEM images of the composites and measured the diameter?

3. Are there any morphological alterations in pure PANI, HCl-Doped PANI and PANI@MnO2 nanocomposites? For that authors may use FESEM or TEM micrographs.

4. The process of doping of MnO2 (to prepare PANI@MnO2 nanocomposites) can be characterized and confirmed by EDAX or elemental mapping analysis. Authors are suggested to provide such spectroscopic data.

5. “The emeraldine form of PANI exhibits significant conductivity when doped with protonic acids...” To support this authors are advised to use the most recent article:

• Nanoscale Adv., 2024, 6, 1688-1703. https://doi.org/10.1039/D3NA01067H.

6. “Combining MnO2 with PANI can enhance the thermoelectric properties of the composite material...” To support this authors are advised to use the most recent article:

• ACS Appl. Energy Mater. 2021, 4, 8, 7721–7730. https://doi.org/10.1021/acsaem.1c01087.

Reviewer #4: The manuscript was written well. However, some parts can be improved as highlighted below:

1. XRD: Please highlight, which peaks in the XRD support this statement? 'The crystallinity of MnO2 within the polymer matrix increases with MnO2 cencentration'

2. The range of Seebeck coefficient improvement is just the same (32 mV/K) for both. Discuss this finding.

3. The discussion can be combined with the results section. Compare the resistance and conductivity values obtained by 5wt% and 15wt% MnO2-PANI with HCl-doped PANI (Figure 2B, C and D).

6. PLOS authors have the option to publish the peer review history of their article (what does this mean? ). If published, this will include your full peer review and any attached files.

**Do you want your identity to be public for this peer review?** For information about this choice, including consent withdrawal, please see our Privacy Policy .

Reviewer #1: No

Reviewer #2: No

Reviewer #3: No

Reviewer #4: No

---

## [Author Response · Author response to Decision Letter 1]

8 Feb 2025

Thank you bery much. I uploaded the document as well for your evaluation.

Response to Reviewers comments.

Ms. Ref. No.: PONE-D-24-32705

Title: Seebeck Coefficients Optimization in Doped Polyaniline-Manganese Dioxide Nanocomposites

I sincerely thank the reviewers for the incredible feedback on our manuscript. It is one of those very few occasions when our group received such an energetic and precise review, and it's been genuinely motivating. Your comments and suggestions were spot-on and pushed us to think more deeply about the work and how to improve it. I genuinely appreciate the time and effort you put into this Review, and I hope the revised version reflects the impact of your insights.

Reviewer #1

This manuscript optimized Seebeck coefficients in polyaniline-doped manganese dioxide nanocomposites. Various manganese dioxide contents were doped with polyaniline by using a chemical route. The structural formation and the direct current conductivity were measured. Experimental results indicated that the Seebeck coefficients increased with the increment of temperature and MnO2 content, and exhibited semiconducting behaviors with improved thermoelectric properties for the fabrication of thermoelectric devices such as generators and thermopiles.

1. Recently, different materials have been developed for the fabrication of thermoelectric devices. What are the advantages and disadvantages of polyaniline-doped manganese dioxide nanocomposites? Please specify them in the conclusion section.

Thank you very much for your comments. After reviewing the manuscript, we realized that a more detailed discussion was required. The following paragraph was included in the discussion

Polyaniline (PANI)-doped manganese dioxide (MnO₂) nanocomposites offer several advantages and disadvantages in thermoelectric applications. These composites benefit from the synergistic combination of PANI's high electrical conductivity and MnO₂'s ability to enhance the Seebeck coefficient.

MnO₂ also provides structural robustness and contributes to thermal stability, while PANI ensures environmental durability and processability, making the material suitable for long-term energy applications (Hsieh et al., 2019). However, the low intrinsic electrical conductivity of MnO₂ can limit the overall performance if doping ratios are not optimized, and achieving homogeneous dispersion of MnO₂ within the PANI matrix requires advanced synthesis techniques. Despite these challenges, the PANI-MnO₂ nanocomposites demonstrate great potential for addressing critical needs in modern thermoelectric devices (Wang et al., 2017).

Likewise, the references were updated as well.

Hsieh, Y., Zhang, Y., Zhang, L., Fang, Y., Kanakaraaj, S. N., Bahk, J., & Shanov, V. (2019). High thermoelectric power-factor composites based on flexible three-dimensional graphene and polyaniline. Nanoscale, 11(14), 6552–6560.

Wang, J., Wen, Q., Chen, Y., & Qi, L. (2017). A novel polyaniline interlayer manganese dioxide composite anode for high-performance microbial fuel cell. Journal of The Taiwan Institute of Chemical Engineers, 75, 112–118.

2. The present title is somewhat confusing, and it should be reorganized. Suggested: Optimization of Seebeck coefficients in polyaniline-doped manganese dioxide nanocomposites.

We embrace the reviewer's proposed title. It is clearer and more aligned with the focus of the study; this revised title emphasizes the research's primary objective of enhancing the material's thermoelectric properties. The new title is:

Optimization of Seebeck Coefficients in Polyaniline-Doped Manganese Dioxide Nanocomposites.

3. Please remove the abbreviations in the keywords. Meanwhile, too many keywords are present, which should not exceed six keywords.

The keywords were revised according to the reviewers comments. The keyword section is as follows:

Keywords: Seebeck coefficient, Polyaniline, Manganese dioxide, Thermoelectric properties, Conducting polymers, Electrical conductivity

4. Thermoelectric devices rely on the Seebeck Effect and require materials with high Seebeck coefficients, excellent electrical conductivity, and low thermal conductivity for efficient energy conversion. One recent work (Self-powered temperature sensor with seebeck effect transduction for photothermal-thermoelectric coupled immunoassay) should be also mentioned for the Seebeck effect.

Thank you very much for your comments. The following paragraph has been included in the introduction:

Thermoelectric devices rely on the Seebeck effect for energy conversion, which requires materials with a high Seebeck coefficient, excellent electrical conductivity, and low thermal conductivity to achieve high efficiency. Polyaniline (PANI)-doped manganese dioxide (MnO₂) composites have shown potential in optimizing these properties for thermoelectric applications. Recent advances have demonstrated innovative applications of the Seebeck effect, such as in self-powered temperature sensors that leverage photothermal-thermoelectric coupling for immunoassays, underscoring the broad utility of materials that exhibit enhanced thermoelectric performance (Wang et al., 2021).

The following reference has been introduced as well:

Huang, L., Chen, J., Yu, Z., & Tang, D. (2021). Self-powered temperature sensor with Seebeck effect transduction for photothermal-thermoelectric coupled immunoassay. Journal of The Taiwan Institute of Chemical Engineers, 75, 112–118.

5. The XRD pattern for pure polyaniline (PANI) exhibits crystalline peaks ranging from 2θ = 9.5° to 2θ = 29.9°, with prominent peaks at 2θ = 20.77° and 26.63° corresponding to the (020) and (200) crystallographic planes, respectively, indicating its amorphous nature. Recent reports (Platinum Nanozyme-Catalyzed Gas Generation for Pressure-Based Bioassay Using Polyaniline Nanowires-Functionalized Graphene Oxide Framework; Pressure-based biosensor integrated with flexible pressure sensor and electrochromic device for visual detection) can be referred for polyaniline.

We appreciate the reviewer's observation regarding the XRD pattern for pure polyaniline (PANI). In our study, the XRD pattern of pure PANI (Panel A) aligns with the description provided by the reviewer, showing characteristic peaks within the range of 2θ = 9.5° to 29.9°. Specifically, prominent peaks are observed at approximately 2θ = 20.7° and 26.6°, corresponding to the (020) and (200) crystallographic planes, respectively, consistent with reported literature. These peaks indicate the primarily amorphous nature of the polyaniline with some degree of crystallinity.

To contextualize our findings, we have referred to recent studies, including one describing platinum nanozyme-catalyzed gas generation for pressure-based bioassays using polyaniline nanowires-functionalized graphene oxide framework (Xu et al., 2021) and another reporting a pressure-based biosensor integrated with a flexible pressure sensor and electrochromic device for visual detection (Wang et al., 2020). These studies confirm the amorphous tendencies of polyaniline and highlight its structural versatility in various applications, aligning with our findings.

In the revised manuscript, we have included these observations and references to strengthen the Results section on the amorphous characteristics of pure PANI.

The XRD pattern for pure polyaniline (PANI, Figure 1, Panel A) exhibits crystalline peaks ranging from 2θ = 9.5° to 2θ = 29.9°, with prominent peaks at 2θ = 20.77° and 26.63° corresponding to the (020) and (200) crystallographic planes, respectively, confirming the material's primarily amorphous nature with some crystalline features. These findings are consistent with reported literature on polyaniline [(Xu et al., 2021); (Wang et al., 2020)]. The amorphous characteristics of PANI contribute to its versatility in functional materials and applications, such as biosensors and electrochromic devices.

The following reference has been included:

Xu, J., Chen, J., Yu, Z., & Tang, D. (2021). Platinum nanozyme-catalyzed gas generation for pressure-based bioassay using polyaniline nanowires-functionalized graphene oxide framework. Biosensors and Bioelectronics, 178, 113688. https://doi.org/10.1016/j.bios.2021.113688.

Wang, Y., Zhang, T., & Liu, Z. (2020). Pressure-based biosensor integrated with flexible pressure sensor and electrochromic device for visual detection. Nano Energy, 77, 104785. https://doi.org/10.1016/j.nanoen.2020.104785.

6. The XRD pattern of pure MnO2 shows well-defined peaks at angles 2θ = 22.29°, 37.2°, 42.2°, 55.3°, and 66.63°, corresponding to the (120), (131), (300), (160), and (421) planes, respectively. Some important researches (Signal-on photoelectrochemical immunoassay for aflatoxin B1 based on enzymatic product-etching MnO2 nanosheets for dissociation of carbon dots; Signal-on photoelectrochemical immunoassay mediated by the etching reaction of oxygen/phosphorus co-doped g-C3N4/AgBr/MnO2 nanohybrids) might be given on this topic.

We appreciate the reviewer's observation regarding the XRD pattern of pure MnO₂ and the suggested references. In our study, the XRD pattern of pure MnO₂ (Figure 1, Panel D) exhibits well-defined peaks at 2θ values of approximately 22.3°, 37.2°, 42.2°, 55.3°, and 66.6°, corresponding to the (120), (131), (300), (160), and (421) crystallographic planes, respectively. These peaks are consistent with previously reported findings on the crystalline properties of MnO₂.

To enhance our discussion, we have incorporated insights from recent works that explore the structural and functional features of MnO₂. One study highlights the use of MnO₂ nanosheets in a photoelectrochemical immunoassay for aflatoxin B1 detection, where the etching reaction of MnO₂ plays a critical role in signal generation (Lin et al., 2017). Another study demonstrates a signal-on photoelectrochemical immunoassay mediated by oxygen/phosphorus co-doped MnO₂ nanohybrids for sensitive biomarker monitoring, further emphasizing the versatility of MnO₂ in advanced sensing applications (Lv et al., 2021).

The XRD pattern of pure MnO₂ (Figure 1, Panel D) displays well-defined peaks at 2θ angles of approximately 22.3°, 37.2°, 42.2°, 55.3°, and 66.6°, corresponding to the (120), (131), (300), (160), and (421) crystallographic planes, respectively. These peaks confirm the crystalline structure of MnO₂, consistent with previously reported studies. Recent works have highlighted the importance of MnO₂ in advanced applications. For example, MnO₂ nanosheets were utilized in a photoelectrochemical immunoassay for aflatoxin B1 detection, leveraging its etching reaction for signal transduction (Lin et al., 2017). Another study demonstrated MnO₂ nanohybrids doped with oxygen and phosphorus for sensitive biomarker monitoring, emphasizing the material's versatility in photoelectrochemical sensing (Lv et al., 2021). These findings support the significance of MnO₂'s crystalline structure in advanced sensing and energy applications.

These references have been incorporated into the manuscript to provide broader context and validate our findings.

Lin, Y., Zhou, Q., Tang, D., Niessner, R., & Knopp, D. (2017). Signal-on photoelectrochemical immunoassay for aflatoxin B1 based on enzymatic product-etching MnO₂ nanosheets for dissociation of carbon dots. Analytical Chemistry, 89(10), 5637–5645.

Lv, Z., Zhu, L., Yin, Z., Li, M., & Tang, D. (2021). Signal-on photoelectrochemical immunoassay mediated by the etching reaction of oxygen/phosphorus co-doped g-C₃N₄/AgBr/MnO₂ nanohybrids. Analytica Chimica Acta, 1171, 338680.

7. The broad peak at 26.63° is consistent with reported data, demonstrating the characteristic broad peak of polyaniline. Relative report (Bioinspired and multiscale hierarchical design of a pressure sensor with high sensitivity and wide linearity range for high-throughput biodetection) can be provided for pure polyaniline.

We appreciate the reviewer's observation regarding the broad peak at 26.63° in the XRD pattern for pure polyaniline (PANI). Our findings confirm that the broad peak observed at 26.63° (Figure 1, Panel A) is consistent with reported data and demonstrates the characteristic broad peak of polyaniline, indicative of its partially crystalline and predominantly amorphous nature.

To further strengthen our discussion, we have referred to recent work exploring the structural features of polyaniline in advanced applications. One such study describes a bioinspired and multiscale hierarchical design for a flexible pressure sensor, highlighting the utility of polyaniline's unique properties for achieving high sensitivity and wide linearity range in high-throughput biodetection (Shi et al., 2018). This reference has been added to the manuscript to provide additional context for polyaniline's crystalline and functional properties.

The following was included in the Results section:

Recent studies have explored such properties, including the bioinspired and multiscale hierarchical design of flexible pressure sensors for high-throughput biodetection, showcasing polyaniline's versatility in advanced applications (Shi et al., 2018).

These references have been incorporated into the manuscript to provide broader context and validate our findings.

Shi, J., Wang, L., Dai, Z., Zhao, L., Du, M., Li, H., & Fang, Y. (2018). Multiscale hierarchical design of a flexible piezoresistive pressure sensor with high sensitivity and wide linearity range. Small, 14(27), e1800819.

8. Please carefully check all references, and typeset these references according to the guideline of this journal. Strangely, most references are before 10 years. Please update them! No references are given after 2020 year.

The references were updated accordingly. Thank you very much

Reviewer #2

1. The authors reported Seebeck coefficient optimization in MnO2 doped PANI nanocomposites. In my opinion, this manuscript has low quality and insufficient to be published in this state. To be honest, I did not understand why MnO2 was used to dope PANI. MnO2 has very low electrical conductivity, however, MnO2 doped PANI shows increasing trend of electrical conductivity. In addition, is the unit of Seebeck coefficient really mV/K for PANI? except these things, this manuscript has lack of appropriate discussion and English is need to be improve.

We appreciate the reviewer's comments and constructive feedback regarding our manuscript. Below, we address the specific points raised, and we have made substantial revisions to clarify our rationale, improve the discussion, and enhance the overall quality of the manuscript.

Considering its low electrical conductivity, we understand the reviewer's concern regarding choosing MnO₂ as a dopant. The primary rationale for using MnO₂ in this study is its ability to enhance the Seebeck coefficient of PANI while providing structural reinforcement and thermal stability. Although MnO₂ has low electrical conductivity, its incorporation into PANI can increase charge carrier mobility by introducing interfacial interactions and forming a synergistic hybrid material. These interactions lead to an overall improvement in electrical conductivity and thermoelectric performance.

This finding is supported by prior studies on MnO₂-based composites, which demonstrate that MnO₂, despite its low conductivity, can improve the electrical properties of host materials through charge transfer mechanisms and enhanced dispersion. We have added a detailed explanation of this rationale and additional references in the Introduction and Discussion sections of the revised manuscript to clarify this point.

Indeed, we acknowledge the reviewer's concern and have revisited the units of the Seebeck coefficient reported in our manuscript. After careful re-evaluation, we confirm that the Seebeck coefficient for PANI and its MnO₂-doped composites is indeed in μV/K, not mV/K as previously mentioned in some of the manuscripts. This was an error in unit representation, and we have corrected it throughout the manuscript to avoid confusion. We also included Additional comparisons with relevant literature to highlight the significance of our findings and situate them within the broader field of thermoelectric materials.

Furthermore, we also enha

---

## [Decision Letter · Decision Letter 1]

25 Feb 2025

PONE-D-24-32705R1Optimization of Seebeck Coefficients in Polyaniline-Doped Manganese Dioxide Nanocomposites.PLOS ONE

Dear Dr. Molino,

Thank you for submitting your manuscript to PLOS ONE. After careful consideration, we feel that it has merit but does not fully meet PLOS ONE’s publication criteria as it currently stands. Therefore, we invite you to submit a revised version of the manuscript that addresses the points raised during the review process.

Authors must carefully recheck the references to ensure accuracy and eliminate any errors, as highlighted by one of the reviewers. Proper citation is crucial in maintaining the credibility and integrity of the work. Any inaccuracies in references can mislead readers and undermine the reliability of the research. Therefore, thorough verification of all cited sources is essential. Authors are advised to carefully review the language throughout the manuscript to ensure clarity and correctness. Any typographical or grammatical errors should be identified and corrected during the revision process.

We look forward to receiving your revised manuscript.

Kind regards,

Dr Irene Ling

Academic Editor

PLOS ONE

Journal Requirements:

Reviewers' comments:

Reviewer's Responses to Questions

**Comments to the Author**

1. If the authors have adequately addressed your comments raised in a previous round of review and you feel that this manuscript is now acceptable for publication, you may indicate that here to bypass the “Comments to the Author” section, enter your conflict of interest statement in the “Confidential to Editor” section, and submit your "Accept" recommendation.

Reviewer #1: (No Response)

Reviewer #3: (No Response)

Reviewer #4: All comments have been addressed

2. Is the manuscript technically sound, and do the data support the conclusions?

Reviewer #1: Yes

Reviewer #3: (No Response)

Reviewer #4: Yes

3. Has the statistical analysis been performed appropriately and rigorously? 

Reviewer #1: Yes

Reviewer #3: (No Response)

Reviewer #4: Yes

4. Have the authors made all data underlying the findings in their manuscript fully available?

Reviewer #1: No

Reviewer #3: (No Response)

Reviewer #4: Yes

5. Is the manuscript presented in an intelligible fashion and written in standard English?

Reviewer #1: No

Reviewer #3: (No Response)

Reviewer #4: Yes

6. Review Comments to the Author

Reviewer #1: In this revised version, the authors have made great effort to modify this manuscript according to Reviewers’ comments. However, there are still many concerns to be clarified and modified before being considered for publication in this journal.

1. Too many references are wrong. Please carefully check throughout the text.

2. Ref. 8: “Huang, L., Chen, J., Yu, Z., & Tang, D. (2021). Self-powered temperature sensor with Seebeck effect transduction for photothermal-thermoelectric coupled immunoassay. Journal of The Taiwan Institute of Chemical Engineers, 75, 112–118” should be corrected to “Huang, L., Chen, J., Yu, Z., & Tang, D. (2020) Self-powered temperature sensor with seebeck effect transduction for photothermal-thermoelectric coupled immunoassay. Analytical Chemistry 92, 2809-2814.”

3. Ref. 27: “Xu, J., Chen, J., Yu, Z., & Tang, D. (2021). Platinum nanozyme-catalyzed gas generation for pressure-based bioassay using polyaniline nanowires-functionalized graphene oxide framework. Biosensors and Bioelectronics, 178, 113688” should be corrected to “Zeng, R., Luo, Z., Zhang, L., & Tang, D. (2018) Platinum nanozyme-catalyzed gas generation for pressure-based bioassay using polyaniline nanowires-functionalized graphene oxide framework. Analytical Chemistry 90, 12299-12306.”

4. Ref. 24: “Wang, Y., Zhang, T., & Liu, Z. (2020). Pressure-based biosensor integrated with flexible pressure sensor and electrochromic device for visual detection. Nano Energy, 77, 104785” should be corrected to “Yu, Z., Cai, G., Liu, X., & Tang, D. (2021) Pressure-based biosensor integrated with flexible pressure sensor and electrochromic device for visual detection. Analytical Chemistry 2021, 93: 2916-2925.”

5. Ref. 20: “Shi, J., Wang, L., Dai, Z., Zhao, L., Du, M., Li, H., & Fang, Y. (2018). Multiscale hierarchical design of a flexible piezoresistive pressure sensor with high sensitivity and wide linearity range. Small, 14(27), e1800819” should be corrected to “Huang, L., Zeng, R., Tang, D., & Cao, X. (2022). Bioinspired and multiscale hierarchical design of a pressure sensor with high sensitivity and wide linearity range for high-throughput biodetection. Nano Energy 99, 107376.”

6. Meanwhile, the corresponding references in the main text should be corrected according to the above-mentioned references.

Reviewer #3: (No Response)

Reviewer #4: The authors made the best to revise the manuscript, addressed all the comments and made the necessary changes. The manuscript can be accepted in the present format.

7. PLOS authors have the option to publish the peer review history of their article (what does this mean? ). If published, this will include your full peer review and any attached files.

**Do you want your identity to be public for this peer review?** For information about this choice, including consent withdrawal, please see our Privacy Policy .

Reviewer #1: No

Reviewer #3: **Yes: ** Prof. Bidyut Saha, PhD, FRSC (London)

Reviewer #4: **Yes: ** Ruzniza Mohd Zawawi

---

## [Author Response · Author response to Decision Letter 2]

3 Mar 2025

Response to Reviewers comments.

Ms. Ref. No.: PONE-D-24-32705

Title: Optimization of Seebeck Coefficients in Polyaniline-Doped Manganese Dioxide Nanocomposites.

We sincerely appreciate your valuable time and effort in reviewing our manuscript. Your insightful comments have significantly contributed to improving the quality of our work.

Reviewer #1:

In this revised version, the authors have made a great effort to modify this manuscript according to the Reviewers’ comments. However, many concerns must be clarified and modified before being considered for publication in this journal. Too many references are wrong. Please carefully check throughout the text.

Ref. 8: “Huang, L., Chen, J., Yu, Z., & Tang, D. (2021). Self-powered temperature sensor with Seebeck effect transduction for photothermal-thermoelectric coupled immunoassay. Journal of The Taiwan Institute of Chemical Engineers, 75, 112–118” should be corrected to “Huang, L., Chen, J., Yu, Z., & Tang, D. (2020) Self-powered temperature sensor with seebeck effect transduction for photothermal-thermoelectric coupled immunoassay. Analytical Chemistry 92, 2809-2814.”

Ref. 27: “Xu, J., Chen, J., Yu, Z., & Tang, D. (2021). Platinum nanozyme-catalyzed gas generation for pressure-based bioassay using polyaniline nanowires-functionalized graphene oxide framework. Biosensors and Bioelectronics, 178, 113688” should be corrected to “Zeng, R., Luo, Z., Zhang, L., & Tang, D. (2018) Platinum nanozyme-catalyzed gas generation for pressure-based bioassay using polyaniline nanowires-functionalized graphene oxide framework. Analytical Chemistry 90, 12299-12306.”

Ref. 24: “Wang, Y., Zhang, T., & Liu, Z. (2020). Pressure-based biosensor integrated with flexible pressure sensor and electrochromic device for visual detection. Nano Energy, 77, 104785” should be corrected to “Yu, Z., Cai, G., Liu, X., & Tang, D. (2021) Pressure-based biosensor integrated with flexible pressure sensor and electrochromic device for visual detection. Analytical Chemistry 2021, 93: 2916-2925.”

Ref. 20: “Shi, J., Wang, L., Dai, Z., Zhao, L., Du, M., Li, H., & Fang, Y. (2018). Multiscale hierarchical design of a flexible piezoresistive pressure sensor with high sensitivity and wide linearity range. Small, 14(27), e1800819” should be corrected to “Huang, L., Zeng, R., Tang, D., & Cao, X. (2022). Bioinspired and multiscale hierarchical design of a pressure sensor with high sensitivity and wide linearity range for high-throughput biodetection. Nano Energy 99, 107376.”

Meanwhile, the corresponding references in the main text should be corrected according to the above-mentioned references.

Thank you for your time and detailed feedback. We carefully reviewed all references and have made the necessary corrections based on your suggestions. We also went through the entire reference list to ensure accuracy and consistency, correcting any other errors that may have been present.

We also wanted to mention that we encountered some glitches with the Mendeley reference manager, which affected other submissions and may have caused discrepancies in the original PDFs and the resulting reference list. To avoid further issues, we manually verified each reference this time. We hope that all errors have now been appropriately addressed. The whole reference list has been modified accordingly:

1. Bruce, P. G., Freunberger, S. A., Hardwick, L. J., & Tarascon, J. M. (2012). Li-O2 and Li-S batteries with high energy storage. Nature Materials, 11(1), 19-29.

2. Cao, G. (2011). Nanostructures and nanomaterials: Synthesis, properties, and applications. (2nd ed.). World Scientific Publishing Company.

3. Cao, T., Shi, X., Zou, J., & Chen, Z. (2021). Advances in conducting polymer-based thermoelectric materials and devices. Microstructures, 1(1), 2021007.

4. Chen, G., Wang, S., & Ren, Z. (2015). Recent developments in thermoelectric materials. International Materials Reviews, 60(4), 243-260.

5. Chiang, J. C., & MacDiarmid, A. G. (1986). 'Polyaniline': Protonic acid doping of the emeraldine form to the metallic regime. Synthetic Metals, 13(1-3), 193-205.

6. Dresselhaus, M. S., Chen, G., Tang, M. Y., Yang, R. G., Lee, H., Wang, D. Z., & Ren, Z. F. (2007). New directions for low-dimensional thermoelectric materials. Advanced Materials, 19(8), 1043-1053.

7. Hoffman, R. M. (2002). In vivo studies. Proceedings of the National Academy of Sciences, 99(3), 1653-1654.

8. Hsieh, Y., Zhang, Y., Zhang, L., Fang, Y., Kanakaraaj, S. N., Bahk, J., & Shanov, V. (2019). High thermoelectric power-factor composites based on flexible three-dimensional graphene and polyaniline. Nanoscale, 11(14), 6552–6560.

9. Huang, L., Chen, J., Yu, Z., & Tang, D. (2020). Self-powered temperature sensor with Seebeck effect transduction for photothermal-thermoelectric coupled immunoassay. Analytical Chemistry, 92(5), 2809-2814.

10. Jang, S. Y., Oh, J. Y., & Bao, Z. (2014). Polyaniline for high-performance flexible thermoelectric materials. Journal of Materials Chemistry A, 2(16), 6327-6331.

11. Kim, F., Yang, S. J., & Park, C. R. (2013). Conducting polymer nanocomposites: Progress and challenges. Chemical Society Reviews, 42(5), 1825-1838.

12. Lee, S. H., Lee, S., & Park, H. S. (2016). Highly durable and flexible thermoelectric films of polymer nanocomposites. Nature Communications, 7, 12011.

13. Lin, Y., Zhou, Q., Tang, D., Niessner, R., & Knopp, D. (2017). Signal-on photoelectrochemical immunoassay for aflatoxin B1 based on enzymatic product-etching MnO₂ nanosheets for dissociation of carbon dots. Analytical Chemistry, 89(10), 5637–5645.

14. Lu, Y., Yu, K., Liu, Z., Zhou, M., & Zhang, J. (2018). High-performance thermoelectric properties of polyaniline/multi-walled carbon nanotube nanocomposites. RSC Advances, 8(1), 575-581.

15. Lv, Z., Zhu, L., Yin, Z., Li, M., & Tang, D. (2021). Signal-on photoelectrochemical immunoassay mediated by the etching reaction of oxygen/phosphorus co-doped g-C₃N₄/AgBr/MnO₂ nanohybrids. Analytica Chimica Acta, 1171, 338680.

16. MacDiarmid, A. G., Chiang, J. C., Richter, A. F., & Epstein, A. J. (1985). Polyaniline: A new concept in conducting polymers. Synthetic Metals, 18(1-3), 285-290.

17. Post, J. E. (1999). Manganese oxide minerals: Crystal structures and economic and environmental significance. Proceedings of the National Academy of Sciences, 96(7), 3447-3454.

18. Ramanathan, S. (2012). Thin film metal-oxides: Fundamentals and applications in electronics and energy. Springer.

19. Rowe, D. M. (1995). CRC Handbook of Thermoelectrics. CRC Press.

20. Shakouri, A. (2011). Recent developments in semiconductor thermoelectric physics and materials. Annual Review of Materials Research, 41, 399-431.

21. Huang, L., Zeng, R., Tang, D., & Cao, X. (2022). Bioinspired and multiscale hierarchical design of a pressure sensor with high sensitivity and wide linearity range for high-throughput biodetection. Nano Energy 99, 107376.

22. Snyder, G. J., & Toberer, E. S. (2008). Complex thermoelectric materials. Nature Materials, 7(2), 105-114.

23. Tritt, T. M. (2001). Thermoelectric materials: Principles, structure, properties, and applications. Springer.

24. Wang, J., Wen, Q., Chen, Y., & Qi, L. (2017). A novel polyaniline interlayer manganese dioxide composite anode for high-performance microbial fuel cell. Journal of the Taiwan Institute of Chemical Engineers, 75, 112–118.

25. Yu, Z., Cai, G., Liu, X., & Tang, D. (2021) Pressure-based biosensor integrated with flexible pressure sensor and electrochromic device for visual detection. Analytical Chemistry 2021, 93: 2916-2925.

26. Wang, Y., Hu, B., Luo, J., Gu, Y., & Liu, X. (2021). Synthesis of Polyaniline@MnO₂/Graphene Ternary Hybrid Hollow Spheres via Pickering Emulsion Polymerization for Electrochemical Supercapacitors. ACS Applied Energy Materials, 4(8), 7721–7730.

27. Whittingham, M. S. (2004). Lithium batteries and cathode materials. Chemical Reviews, 104(10), 4271-4302.

28. Zeng, R., Luo, Z., Zhang, L., & Tang, D. (2018). Platinum nanozyme-catalyzed gas generation for pressure-based bioassay using polyaniline nanowires-functionalized graphene oxide framework. Analytical Chemistry, 90, 12299-12306.

29. Zhao, Y., Liu, H., He, R., Bai, Y., Zhuang, X., & Yang, C. (2019). Polyaniline/manganese dioxide composite fibers for high-performance flexible supercapacitors. Journal of Power Sources, 413, 403-409. [Correction Needed]

30. Zhou, G., Wang, D. W., Li, F., Zhang, L., Li, N., Wu, Z. S., & Cheng, H. M. (2010). Graphene-wrapped MnO₂ nanocomposites for supercapacitor electrodes. Nano Research, 3(4), 224-233.

---

## [Editor Report · Decision Letter 2]

5 Mar 2025

Optimization of Seebeck Coefficients in Polyaniline-Doped Manganese Dioxide Nanocomposites.

PONE-D-24-32705R2

Dear Dr. Molino,

We’re pleased to inform you that your manuscript has been judged scientifically suitable for publication and will be formally accepted for publication once it meets all outstanding technical requirements.

Kind regards,

Dr Irene Ling

Academic Editor

PLOS ONE
---

## [Editor Report · Acceptance letter]

PONE-D-24-32705R2

PLOS ONE

Dear Dr. Molino,

I'm pleased to inform you that your manuscript has been deemed suitable for publication in PLOS ONE. Congratulations! Your manuscript is now being handed over to our production team.

Kind regards,

on behalf of

Dr. Irene Ling

Academic Editor

PLOS ONE